# Computer Navigation and 3D Printing in the Surgical Management of Bone Sarcoma

**DOI:** 10.3390/cells10020195

**Published:** 2021-01-20

**Authors:** Robert Allan McCulloch, Tommaso Frisoni, Vineet Kurunskal, Davide Maria Donati, Lee Jeys

**Affiliations:** 1Oncology Arthroplasty Fellow, Royal Orthopaedic Hospital, Birmingham B15 2TT, UK; 23rd Orthopaedic and Traumatologic Clinic prevalently, Oncologic—IRCCS Rizzoli Orthopaedic Institute, 40126 Bologna, Italy; tommaso.frisoni@ior.it (T.F.); Davide.mariadonati@ior.it (D.M.D.); 3Biomedical and Neuromotor Sciences Department, University of Bologna, 40126 Bologna, Italy; 4Orthopaedic Oncology Surgeon, Royal Orthopaedic Hospital, Birmingham B15 2TT, UK; vineet.kurunskal@nhs.net (V.K.); lee.jeys@nhs.net (L.J.)

**Keywords:** osteosarcoma, computer-assisted surgery, patient-specific instrumentation, computer navigation

## Abstract

The long-term outcomes of osteosarcoma have improved; however, patients with metastases, recurrence or axial disease continue to have a poor prognosis. Computer navigation in surgery is becoming ever more commonplace, and the proposed advantages, including precision during surgery, is particularly applicable to the field of orthopaedic oncology and challenging areas such as the axial skeleton. Within this article, we provide an overview of the field of computer navigation and computer-assisted tumour surgery (CATS), in particular its relevance to the surgical management of osteosarcoma.

## 1. Introduction

Osteosarcoma (OS) is defined by neoplasms that have the histological characteristics of producing an osteoid and being associated with malignant mesenchymal cells [1]. There are varying histological subtypes, but the vast majority (80–90%) are high-grade OS [2]. In the majority of cases the aetiology is unknown; however, two genetic mutations are associated with OS: a mutation of the retinoblastoma gene, and an autosomic recessive mutation of p53 in Li–Fraumeni syndrome [3].

The overall incidence of OS is 0.2–3/100,000 per year in Europe and its most common anatomical sites are the distal femur, proximal tibia and proximal humerus [4,5,6]. Ten percent of cases present in the axial skeleton, with the majority originating in the pelvis. This patient subgroup has a higher proportion of patients ages 60 and above [7,8].

Survivorship of OS in extremity localised, non-metastatic disease is 60–70%. However, in metastatic disease, the 5-year overall survival drops to 28–33% [9]. Complete surgical resection remains essential for cure [10], yet there remains some controversy as to the definition of an adequate margin of resection, with evidence that close margins (<5 mm) versus wide margins (>10 mm) did not have an effect on rates of local recurrence [11,12].

Due to the challenges of the surrounding anatomy and the complex three-dimensional structure of the pelvis, a complete resection in the pelvis can be more challenging than in the appendicular skeleton. It has been reported that the probability of an experienced surgeon achieving a 1 cm surgical margin in all three planes on a simulated tumour model of the pelvis was only 52% [13]. The sequelae of inadequate resection are significant, with recurrence rates of 70% and 92% for marginal and intralesional resections, respectively, for OS [14,15].

The uptake in the use of computer navigation in the surgical management of OS is in response to these factors, in a bid to minimise the incidence of inadequate resection margins.

## 2. What Is Computer Navigation?

Computer-navigated surgery has been used in neurosurgery since the 1980s, where improved accuracy in resections for cancer was achieved by mapping brain tumours pre-operatively and using this to plan surgical resection. It is only at the turn of the 21st century that it has been adopted within the orthopaedic community [16,17], principally in the field of spinal surgery. A technology that provides real-time feedback within a field that has a small margin for error has an obvious home in the specialty of orthopaedic oncology.

Computer navigation encompasses all techniques using computing to augment surgical procedures. The two main types of navigation currently used in orthopaedic surgery are “image-based” and “patient-specific instrumentation and reconstruction”.

### 2.1. Patient-Specific Instrumentation and Reconstruction

In recent years, CATS has been developed along with 3D printing technology, which represents the possibility to personalize reconstruction with custom-made prostheses [18,19,20] and improve accuracy in bone cutting using patient-specific instruments (PSI) [21,22].

Manufacturing of 3D-printed PSI jigs and custom-made prostheses is based upon the principle of rapid prototyping (RP). RP is a process that directly produces a physical object with a defined structure and shape on the basis of virtual/mathematical model data. Rapid prototyping was first used in the late 1980s and was developed to apply the precision and functionality of computer-assisted design (CAD) to manufacturing. With this technology, a prototype could be quickly produced and accurately represents the engineers draft [23,24]. Electron beam melting (EMB) is the optimal technology to fabricate metallic components with complex shapes and porous structures and has a pivotal role either in the PSI technology or prosthesis manufacturing. EBM involves the generation of an electron beam focused on a powder layer that is added to a previous one. Each layer (50–100 µm thickness) is pre-heated, and using an electron beam a high temperature is generated (up to 600–800 °C), allowing the fusion of the powder according to the 3D CAD project. This process is repeated layer by layer, building the model in a vacuum chamber [25].

Traditional methods of manufacturing orthopaedic implants used subtractive machining, where the material is removed from a metal block until the planned shape has been achieved. The process of RP is different to this and uses the process of additive manufacturing. This is when a construct is made based upon a digital model by plastic or metal being deposited in layers. This provides versatility when constructing complex geometric shapes. Changes in design require no new equipment; therefore, it is ideally suited to low volume patient-specific instrumentation or implants [25,26,27].

In surgery, 3D printing is utilised in the production of the following [26,27]:Anatomical models.Patient-specific instrumentation (PSI).Custom-made implants and devices.

### 2.2. Anatomical Models

Anatomical model fabrication is the most common use of 3D printing in surgery.

In the planning stage, 3D models are used to represent the anatomy, simulate surgical procedure and test surgical tools (Figure 1 and Figure 2). Studying a patient’s anatomy with a replication of the structures gives a better comprehension compared to 2D images on a computer screen [28,29]. The 3D-printed model can be shown to patients in order to explain the pathology, surgical planning and assist the consenting process [30,31].

### 2.3. PSI (Patient-Specific Instrumentation)

PSI technology is the most popular medical application of 3D printing and has been developed as an alternative to intraoperative navigation. PSI is designed on a 3D model based on CT and MRI fusion according to the surgical approach, desired resection margins and reconstruction method. The unique anatomy of the patient and the PSI shape allows to place the guide only in a pre-defined position. These contact surfaces must be defined by both surgeon and engineer, considering the surgical approach, the bone exposure and tumour extension. These features make the application of PSI more useful for bone deformities and tumours where the anatomy is often abnormal. The final PSI is commonly printed in nylon or polyamide and provided to the surgeon sterile or to be sterilized (Figure 3). Thus, this method reproduces surgical planning with more accuracy than a free-hand approach. Furthermore, it encourages surgeons to perform a “virtual” surgery in advance, leading to a better comprehension of the possible pitfalls and to a tailored treatment for each patient. This complex procedure in some way has changed surgeons’ mindset due to the network that must be established with a team of engineers.

Limb salvage surgery has become the rule for most patients with bone tumours. However, resection of these tumours needs to achieve wide margins to limit local recurrence and at the same time preserve function [32,33]. In pelvic tumours, surgery is more difficult due to the complex three-dimensional (3D) anatomy of the pelvic bone and the presence of neurovascular structures as well as viscera. This situation leads to a higher local recurrence and complication rate in the pelvis compared with long bones [21,34,35]. In long bones, the use of PSI is most useful in joint preserving resection, in particular in children or young patients in which a reconstruction with a bone allograft can be carried out [32,33].

The performance and accuracy of PSI has been demonstrated; it improves alignment and surgical and operating theatre times [36,37,38,39,40,41,42,43], as well as reducing the risk of contamination and wasting of instrument trays, even if some trials have not displayed significant advantages [44]. Nonetheless, in tumour surgery, the possibility to reduce surgical time whilst performing a more precise cut on the bone and reproducing it on an allograft (or a prosthesis) may improve the outcome of the oncological patient [45].

### 2.4. Custom-Made Implants and Devices

Custom-made implants have been used since the 1950s, when a complex reconstruction was performed in a limb salvage procedure [46]. The physical properties of titanium alloy are the most adaptable to bone defect reconstruction. Titanium has high mechanical resistance similar to stainless steel, a low density (50% less than stainless steel) and a low elasticity Young’s modulus, giving the necessary flexibility associated with a high mechanical capacity [47,48,49]. Furthermore, the superficial part of titanium is covered by a free oxygen layer that increase the biocompatibility and biological integration with the surrounding tissues [50,51,52,53].

Many studies have shown that the porosity of titanium is one of the most relevant features influencing bone ingrowth; Frosch et al. has demonstrated that 300–600 µm is the better porosity for osteoconduction [54].

A more recent development is a particular configuration of titanium called trabecular titanium; the main advantage of the trabecular structure is the rough surface that amplifies the contact with host bone, increasing the integration.

## 3. Process

### 3.1. Preoperative Images

A good resolution of pre-operative images is needed in order to have the most accurate representation of the patient-specific anatomy and pathology. CT and MRI scans are essential to approach a complex bone tumour surgery.

### 3.2. Model

A 3D bone-tumour model was created to allow the surgeon to virtually perform the bone resection cuts of the tumour and then to design the cutting guides as well as prosthesis.

To obtain the 3D model, DICOM files (digital imaging and communications in medicine) were loaded into dedicated CAD software (computer-aided design) that allows segmentation and creation of STL files (surface triangulation language).

In this phase, cooperation is needed between the surgeon and engineer in merging the MRI–CT scan images, selecting the tumour/defect area and setting the osteotomy planes.

The STL file can be sent to the 3D printer (or the company that provides the guides or implants) and is the key to achieve the tri-dimensional model.

### 3.3. PSI and Implants

Patient-specific instruments (PSIs) and implants are designed according to the planned resection strategy. The virtual custom prosthesis is created providing optimal fixation and reconstruction of the local osseous anatomy.

The prosthesis is provided with tools to allow fixation to the host bone as well as plates or rods, while the screw positions and lengths were previously planned as well.

For articular reconstruction, a surface layer (polycaprolactone, PCL, or poly-lactic acid, PLA) can be integrated in the titanium porous structure.

The implant is fabricated through the deposition of few micron layers of titanium powder melted by electron beam technology, as previously described. The implant’s porosity is widely variable in order to increase its performance, such as the macro-porosity of body, to lighten the implant, till micro-porosity at the bone surface contact. These surfaces have pores with an average size of 0.7 millimetres, allowing the host bone to grow directly inside the implant spaces to achieve a stable biological fixation (Figure 4A,B).

The major issue of the entire process is the absence of testing of the biomechanical properties. Due to the customization, the product is unique, and validation of the implant is up to the surgeon [55].

## 4. “Image-Based” Surgery

In “image-based” computer-assisted tumour surgery there are three principal areas [56]:-The therapeutic object; the target of treatment, e.g., OS of the distal femur.-The virtual object; the virtual representation of the patients’ anatomy involved in the surgery.-The navigator link; the connection between the two abovementioned objects.

Wong [57] has described three main stages of computer-assisted tumour surgery. Firstly, there is *Preoperative Navigation Planning*. As per PSI planning, CT provides excellent visualisation of osseous anatomy and MRI provides supplementary information regarding soft tissue anatomy. These can be overlaid, combining the strengths of both imaging modalities. Further techniques, such as PET, can be used to help differentiate between malignant and benign tissues and assist in deciding upon resection sites. Once the resection levels have been planned, reconstruction can then be decided upon, whether that be allograft reconstruction or a custom prosthesis [58].

The second stage of CATS is *Intraoperative Execution*. Most commonly in surgery for osteosarcoma, this is done manually using a navigation link. Predefined points on the cross-sectional imaging are matched with the patients’ anatomy. The more points detected the higher the accuracy. Any straight instrument with a pointed tip can be calibrated and tracked with navigation. Once the therapeutic and virtual object have been calibrated, bone resections using instruments such as osteotomes or burrs can then be performed [59].

The final stage of CATS is *Postoperative Validation*; this being validation of the resection performed. Clinical methods for achieving this in an expeditious manner include frozen sections, and due to the dynamic reference markers still being in place once the resection has been completed, one can thus ensure that the resection was completed as planned [60] (Figure 5).

Bosma et al. [61] performed a comparative cadaveric study comparing simulated tumour resections using “image-based”, “PSI” and free-hand techniques. Regarding the accuracy of the planned resection, the free-hand technique was significantly less accurate than either technique or when the “image-based” or “PSI” techniques were combined. Overall, using PSI appeared to be the easiest to use, had the quickest surgical time and best overall accuracy. However, it should be noted that a limitation of this study is that it was based on cadaveric femoral and tibial samples. Therefore, it would not reproduce the potential changes in tumour geometry between the imaging, planned resection and manufacture of the PSI equipment.

## 5. Evidence

The implementation of CATS in the surgical management of OS has been reported in both the appendicular and axial skeleton. However, due to the unique anatomical challenges of the pelvis, it has gained the most traction in this area.

## 6. Axial Skeleton

It is known that surgical margins do predict local recurrence in osteosarcoma [14,15]. A reduction in intralesional resection of 29% to 8.7% in tumours of the sacrum and pelvis following the implementation of CATS was displayed by Jeys et al. [62]. These promising results have been reproduced elsewhere with the work of Bosma et al. [63], where adequate bone margins within pelvic resections for sarcoma was achieved in 50% of cases without the use of navigation, compared to 81% with CATS. Navigated resections can also allow for the preservation of vital structures, allowing for improved patient function (Figure 6).

Abraham reports a series of 23 patients with all cases having a negative bone resection margin but in two patients there was a positive soft tissue margin [64]. This is an important point as navigation is most effective for bony resections and care must be taken regarding a resection of the associated soft tissue mass. This point is reiterated in a series of 23 patients with a sacral or pelvis sarcoma with a mean follow up of 59 months. Despite all patients having clear bone resection margins, there was a local soft tissue recurrence rate of 35.1% at 6 years [65]. Therefore, surgeons must ensure wide margins are prioritised over preservation of function.

Margins of resection using PSI were studied in a comparative series by Docquier et al. [66]. Nine consecutive patients with primary pelvic sarcomas (one osteosarcoma) that underwent resection with PSI were analysed in comparison with a control group of patients previously treated in the same hospital without PSI guidance. All the margins were wide but one. The mean operative time was similar. No local recurrence was reported with a significant statistical difference compared to the previous series.

In the management of pelvic OS, CATS has a role in reconstruction as well as resection. Complication rates in the endoprosthetic reconstruction of pelvic sarcoma is known to be high, with early reports quoting rates of 58.1% [67]. Precision in the fitting of custom prosthetics or allografts is essential in providing longevity to the implant and function to the patient. Jeys presents a series of 33 patients with acetabular resection for sarcoma and reconstruction using an “ice cream cone” prosthesis with and without navigation. The rate of major complication in the navigated and non-navigated series was 9% and 50%, respectively. The navigated cohort had no implant failures. Along with these favourable outcomes from a technical perspective, the navigated patients also had significantly superior functional outcomes using the musculoskeletal tumour society and Toronto extremity salvage scores [68].

Guo et al. [69] reports a series of 80 patients treated for pelvic tumours (12 osteosarcomas) reconstructed with 3D-printed, custom-made implants. In osteosarcoma patients, two local recurrences were observed, leading to a hindquarter amputation. Considering the whole series, 16 patients (of whom 10 had primary bone tumours) died. Infection occurred in 10 patients whilst mechanical complications were observed in 2.

## 7. Appendicular Skeleton

Reconstruction of the appendicular skeleton post OS excision can be with either a prosthesis, allograft or autograft. Wong [59] has published a series of 8 patients managed for sarcoma of the appendicular skeleton managed with computer navigation. The mean age in the case series was 17 years old, and the significance of utilising CATS in this series was that joint preserving surgery was performed in all patients. All had a complete resection with a minimum of 24 months follow up and no evidence of local recurrence. The significance of utilising CATS in this area permits a higher level of accuracy of resection custom prosthetic generation, inserted with higher precision, thus preserving function by maintaining the joint surfaces and the surrounding soft tissues. In 2016, Zhang et al. [32] published a small series of 8 osteosarcoma patients requiring partial resection of distal femur and reconstruction with allograft using 3D-printed guides. The mean follow-up time was 33 months; no recurrence was observed, with a reduction in surgical time and blood loss. Shehadeh [70] describes a series of 31 cases of insertion of a joint-sparing prosthesis using custom implants and patient-specific cutting blocks to guide the resection. All patients had clear bone resection margins but, at a mean follow up of 3 years, 6 patients had a recurrence of local or distant disease. They note an average time of 6 weeks for the manufacture of the jigs and prosthesis once planning from the imaging has been completed, and this delay can be significant in an aggressive fast-growing tumour.

Farfall [71] presents a series of 69 consecutive patients with sarcoma (principally OS) treated with resection and bone allograft reconstruction. Computer navigation was not only used for the resection but also used for surface mapping of the allograft and to guide where the cuts on the graft should be to ensure a good fit with the bone ends. They describe a non-union rate of 6%, which compares well to the literature published on non-CATS allograft surgery [72]. This may well be related to an improvement in the contact area between the native bone and allograft using navigated pre-planned cuts versus freehand osteotomies, which has been suggested previously [73].

The work of Li et al. [74] further supports the utility of CATS in joint preservation, utilising either an allograft or vascularised fibula graft reconstruction. In 9 patients with juxta-articular sarcomas of the lower limb, all managed with the aforementioned technique, all had clear surgical margins with no cases of local recurrence at a mean 25 months follow up. The mean registration error for navigation in the series was 0.40 mm. In a retrospective study, Kolundžić et al. [75] presented a case series in which 11 patients, mostly under 16 years old, were treated for primary bone sarcoma (6 osteosarcomas of the upper limb). Reconstruction was managed with 3D technology in order to improve the functional outcomes and quality of life. All osteosarcoma patients were alive at the last follow-up time; one local recurrence was recorded and two mechanical complications.

## 8. Limitations and Future Developments

Due to the requirement for a link between the therapeutic and virtual object to be made by the surgeon in navigated surgery, inaccuracies in this step can lead to inaccuracy in resection [59]. This risk is minimised with surface matching, and supplementary techniques have been described, including the use of semiautomatic matching and fiducial markers, which involved the insertion of Kirschner wires preoperatively, then CT scanning with the wires in situ to act as reference points [74,76]. It is of vital importance that the imaging used in the planning phase is contemporaneous to prevent a discrepancy between the imaging and the pathology due to tumour progression.

Patient-specific instrumentation does not account for the soft tissue extent of the tumour and, as this is a static, predetermined resection, there is no scope for altering the planning resection intraoperatively unless the PSI jig is discarded. Soft tissues and the soft tissue extent of the tumour may also impact on the jig, sitting correctly on bony landmarks and thereby potentially compromising the resection. Finally, due to the delay between the planning and jig manufacture, there is a possibility for disease progression and therefore a mismatch between the planned resection and tumour margins, resulting in intralesional resections [77].

Surgeon error is also possible, and misplacement of the jigs can again result in intralesional resections [78]. Finally, there is some concern regarding infection risk when using PSI jigs [79].

With the adoption of any new therapeutic technique there is a learning curve attached to it. Computer navigation takes time both preoperatively for planning purposes and also intraoperatively for tracker insertion and registration. Mahendra et al. [76] described the preoperative planning times and intraoperative timings during their adoption of computer navigation. The initial preoperative planning time was 45 min, which decreased to 25 min after the fifth patient in their series. Tracker insertion and registration initially took a mean of 30 min, decreasing to 20 min from the fifth patient onwards. Farfalli et al. [80] published their experience of CAT and found similar results, with the mean time for preoperative planning and registration improving with experience. Their median registration error was 0.6 mm, which did not improve with time. Although their margins for bony resection were wide in all cases, the soft tissue resection was in 20/78 cases. They also had to abandon the use of navigation in 5% of cases due to technical error, albeit all of these occurred during the first 20 cases of the series.

## 9. Conclusions

Computer-assisted tumour surgery offers an innovation that can improve the accuracy of resection in osteosarcoma surgery and thus reduce the risk of recurrence. This is of particular use with complex resection and reconstructions, such as those in the pelvis and periarticular resections of the appendicular skeleton (see Table 1 for overview). Increasing rates of limb salvage in OS mean that the surgeon has to resect closer to the tumour than they would do with an amputation; therefore, the precision provided by CATS is advantageous. CATS has shown benefit with reconstructive methods using both allografts and custom prosthetics. Although there is a learning curve and CATS requires more time for preoperative planning, the intraoperative time has been shown to be reduced.

Although at present the cost of this technology is high [81], with time this will decrease along with diminishing the size of the equipment and the possible use of robotics to further reduce the possibility of surgical error.

Currently published evidence principally relies on case series and basic science; therefore, going forwards, a more robust analysis of this technology and its application to the management of the surgical management of osteosarcoma is necessary.

## Figures and Tables

**Figure 1 cells-10-00195-f001:**
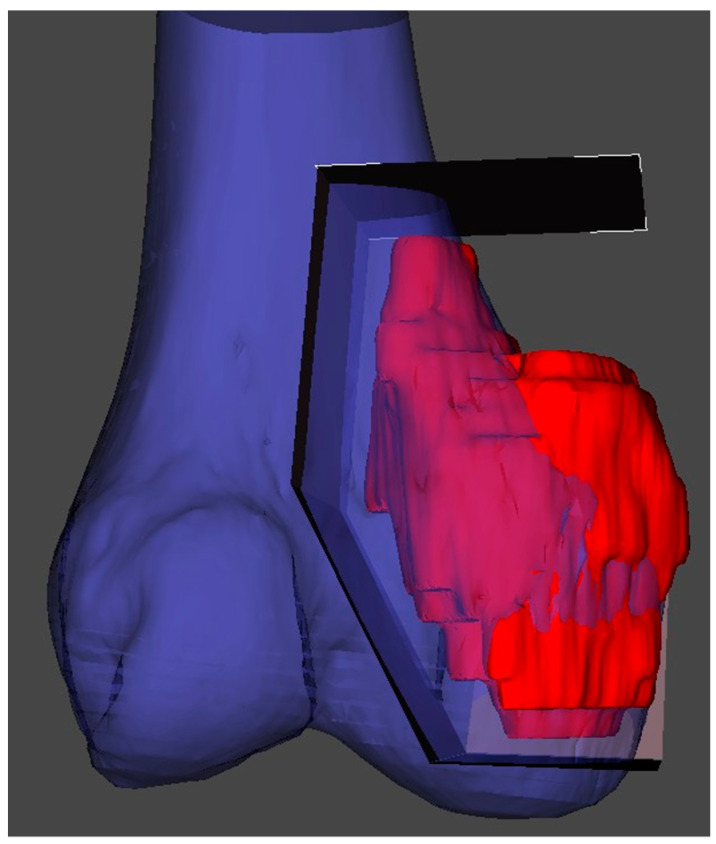
3D reconstruction and planning of a resection for a distal femoral sarcoma.

**Figure 2 cells-10-00195-f002:**
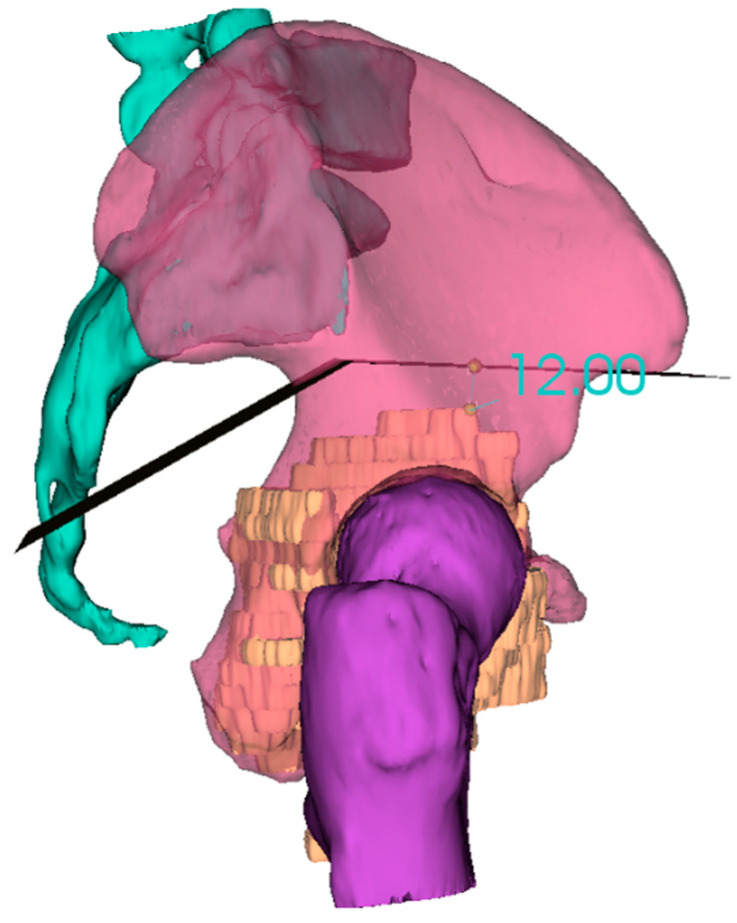
3D reconstruction and planning of a resection for a pelvic sarcoma.

**Figure 3 cells-10-00195-f003:**
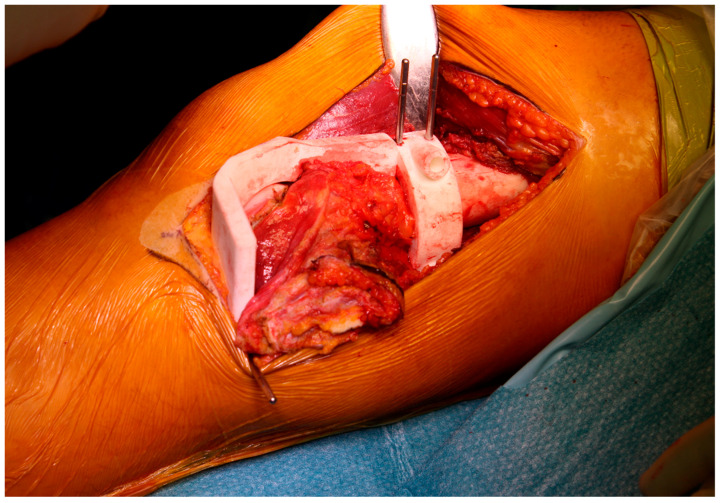
Final patient-specific instrumentation (PSI) printed in nylon and secured intraoperatively.

**Figure 4 cells-10-00195-f004:**
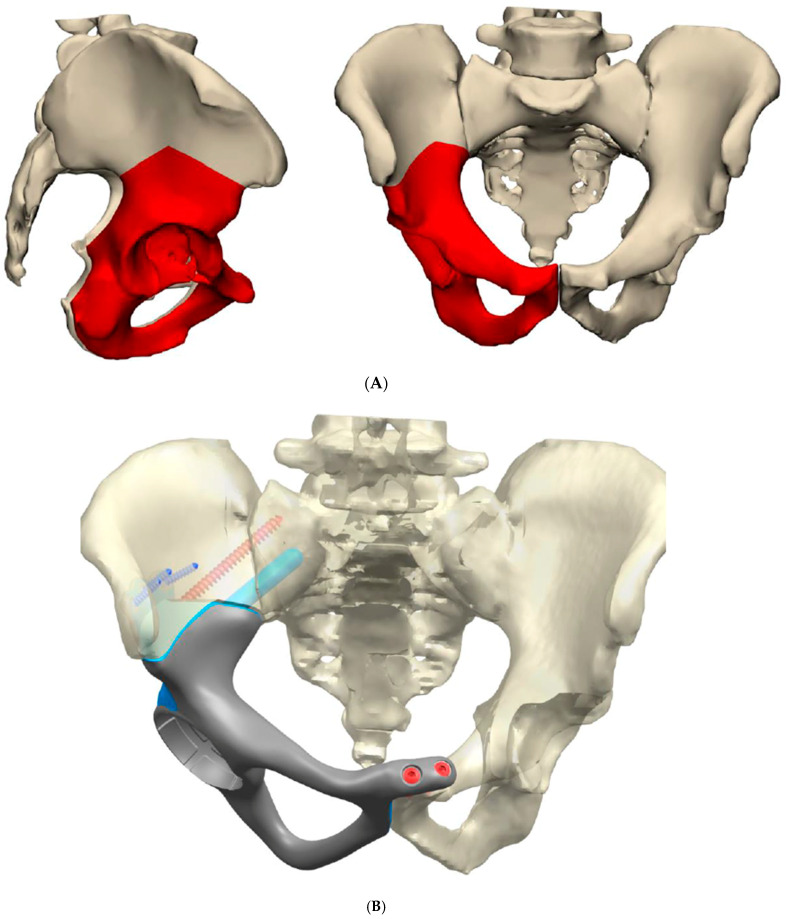
Custom prosthesis post pelvis resection: (**A**) surgical planning using 3D modelling; (**B**) planning for reconstruction post resection; (**C**) comparison between resected pelvic specimen and 3D-printed reconstruction prior to implantation; (**D**) post-operative radiographs.

**Figure 5 cells-10-00195-f005:**
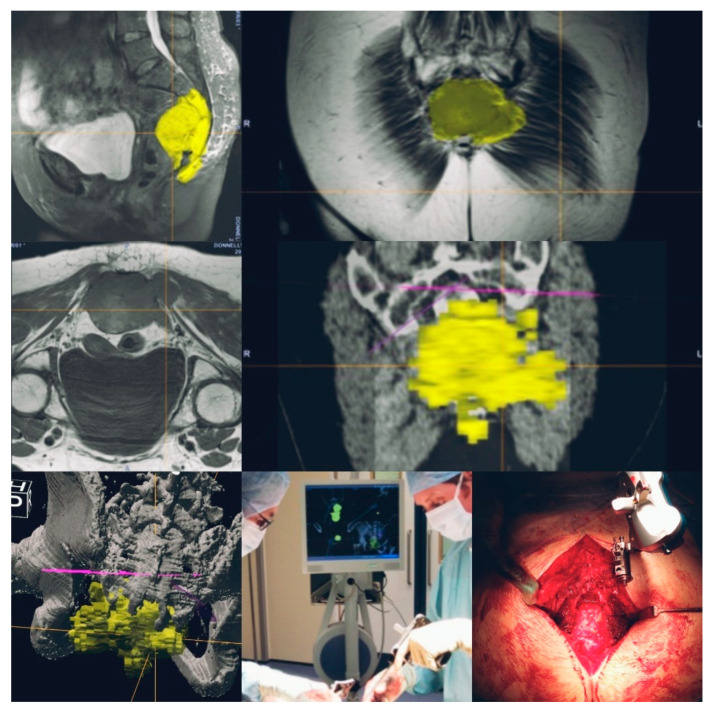
Navigation-assisted sacral sarcoma resection. Sagittal, coronal and axial images showing a lesion in the sacrum (S3 to coccyx) with a presacral mass. Navigation-planned osteotomies (yellow—tumour highlighted; purple line—planned level for osteotomy). Intraoperative navigation-guided resection completed.

**Figure 6 cells-10-00195-f006:**
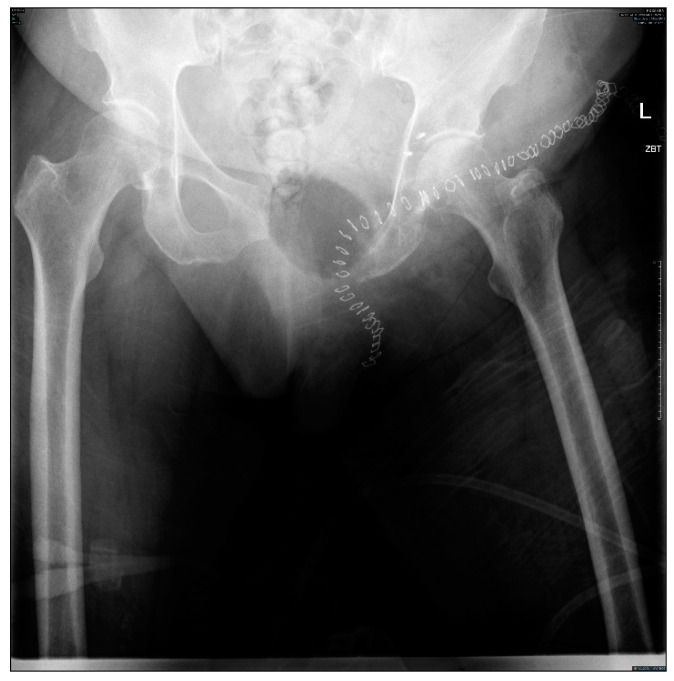
Post-operative radiograph showing the planned navigation resections sparing the native hip joint.

**Table 1 cells-10-00195-t001:** Summary table comparing the “image-based” and “patient-specific instrumentation and reconstruction” methods.

	Image-Based Method	PSI and Reconstruction Method
Clinical workflow.	Pre-operative imaging (CT/MRI) reviewed by the surgical team and resection and reconstruction planning completed using the navigation software.	Collaboration between the surgical and engineering team to confirm the design based upon CT/MRI. Additive manufacturing process and sterilisation.
Intraoperative settings	Navigation equipment required in theatres including sterile equipment trays (Figure 5).	Reduces risk of contamination and wasting of instrument trays [44].
Operative time	Likely longer operative times due to tracker insertion and registration [61].	Potential for decreased operative time [36,37,38,39].
Surgical accuracy	High levels of accuracy compared to freehand techniques [61].	High levels of accuracy compared to freehand techniques but possibility for mismatch between planned resection and tumour margins due to progression [61,77].
Learning Curve	High levels of accuracy from the outset but decreasing operative times with experience [76,80].	Short learning curve but no intraoperative visual feedback if the jigs are placed incorrectly [78].

## Data Availability

Data sharing not applicable No new data were created or analyzed in this study. Data sharing is not applicable to this article.

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
