# Peer review of "Computer Navigation and 3D Printing in the Surgical Management of Bone Sarcoma"

_cells, 2021, doi:10.3390/cells10020195_

Round 1

Reviewer 1 Report

The manuscript represents a well-written comprehensive review of the literature on the state of art of computed assisted tumor surgery. Authors' experience in this field is evident (with some personal clinical cases in the figures), despite no case series is reported. The techniques' description "starts from long aways" to explain their origin and applications also in general orthopaedic surgery, making the text quite long, but it appears necessary due to the complexity of the specific CATS topic.

In my opinion the manuscript appears suitable for publication in the present form as comprehensive literature review.

Author Response

Dear editorial board and reviewer,

many thanks for your comments. As per instruction no changes to the manuscript have been made.

yours sincerely

Robert McCulloch

Reviewer 2 Report

This journal is well written about computer assisted surgery in surgical management of cancer. The contents of this journal is systematic and well organized. But, computer assisted surgery is a widely known method and this paper does not decribe anythiing special. 

In my opinion, the special reason should be describe as to why the authors chose this method in "osteosarcoma" especially. Because we did surgery with CAS in many type of cancer. And the authors should be described prognosis of surgical management with CAS in the authors' hospital or center. 

Author Response

Dear editorial board and reviewer,

many thanks for your comments on our article. Firstly regarding your comment on emphasising the specific application of CATS to OS we have included the following sentence in the conclusion "Increasing rates of limb salvage in OS mean that the surgeon has to resect closer to the tumour than they would do with an amputation therefore the precision provided by CATS is advantageous."

Secondly, within the text we have referenced both institutions experiences of CATS with the following references.

42.Donati D, Di Bella C, Frisoni T, Cevolani L, DeGroot H. Alloprosthetic composite is a suitable reconstruction after periacetabular tumor resection. Clin Orthop Relat Res. 2011 May;469(5):1450-8

65. Jeys L, Matharu GS, Nandra RS, Grimer RJ. Can computer navigation-assisted surgery reduce the risk of an intralesional margin and reduce the rate of local recurrence in patients with a tumour of the pelvis or sacrum? Bone Joint J. 2013 Oct; 95-B(10):1417-24.

68. Nandra R, Matharu G, Stevenson J, Parry M, Grimer R, Jeys L. Long-term outcomes after an initial experience of computer-navigated resection of primary pelvic and sacral bone tumours: soft-tissue margins must be adequate to reduce local recurrences. The bone & joint journal. 2019 Apr;101(4):484-90.

70. Jaiswal PK, Aston WJ, Grimer RJ, Abudu A, Carter S, Blunn G, Briggs TW, Cannon S. Peri-acetabular resection and endoprosthetic reconstruction for tumours of the acetabulum. J Bone Joint Surg Br. 2008 Sep; 90(9):1222-7.

yours sincerely

Robert McCulloch

Reviewer 3 Report

The authors reviewed the current assistive technology (computer navigation and 3D printing) in the surgical management of bone sarcoma.

The title “Computer-assisted surgery in the surgical management of osteosarcoma” is not entirely consistent with the review article. I suggest the title to be “Computer navigation and 3D printing in the surgical management of bone sarcoma.”

In computer-assisted orthopaedic surgery, the term “Imageless Surgery” is used in navigation-assisted arthroplasty without the preoperative imaging. In the imageless navigation, 3D bone models of the patients are created by collecting anatomical data intraoperatively. Therefore, it is confusing if the “imageless Surgery” is now used to represent the 3D printing technique in the review article.

The information on 3D printed implants is insufficient. Authors better describe the advantages of using 3D-printed implants regarding the design freedom and the nature of 3D printing in contrast to the traditional subtractive manufacturing method (1).

A table can be added for easier reference. Computer navigation and 3D printing are compared and contrasted in bone sarcoma surgery. It may include clinical workflow, intraoperative settings, operative time, surgical accuracy, learning curve, limitations, evidence, etc.

Line 131: the unit “300-600” should be a micrometer but not a millimeter.

Reference
1. Wong, KC., Scheinemann, P. Additive manufactured metallic implants for orthopaedic applications. Sci. China Mater. 61, 440–454 (2018). https://doi.org/10.1007/s40843-017-9243-9

Author Response

Dear editorial board and reviewer,

many thanks for your detailed comments. I shall go through them in turn.

Firstly, as suggested we have changed the title to" Computer navigation and 3D printing in the surgical management of bone sarcoma".

Secondly, the ambiguous usage of the term "imageless surgery" has now been replaced to "patient specific instrumentation and reconstruction" throughout the article.

Thirdly, further details have been added regarding the differences between additive and subtractive manufacturing with the following paragraph with appropriate references

"Traditional methods of manufacturing orthopaedic implants used subtractive machining where material is removed from a metal block until the planned shape has been achieved. The process of RP is different to this and uses the process of additive manufacturing. This is when a construct is made based upon a digital model by plastic or metal being deposited in layers. This provides versatility when constructing complex geometric shapes. Changes in design require no new equipment therefore it is ideally suited to low volume patient specific instrumentation or implants 25,26,27."

We have included a summary table (figure 7) providing an overview and comparison of the two methods. 

Line 131 has been corrected to read µm.

yours sincerely

Robert McCulloch